# Oral Supplementation with Hydroxytyrosol Synthesized Using Genetically Modified *Escherichia coli* Strains and Essential Oils Mixture: A Pilot Study on the Safety and Biological Activity

**DOI:** 10.3390/microorganisms11030770

**Published:** 2023-03-17

**Authors:** Yannis V. Simos, Stelios Zerikiotis, Panagiotis Lekkas, Antrea-Maria Athinodorou, Christianna Zachariou, Christina Tzima, Alexandros Assariotakis, Dimitrios Peschos, Konstantinos Tsamis, Maria Halabalaki, Filippos Ververidis, Emmanouil A. Trantas, Garyfalia Economou, Petros Tarantilis, Argyro Vontzalidou, Irini Vallianatou, Charalambos Angelidis, Patra Vezyraki

**Affiliations:** 1Laboratory of Physiology, Department of Medicine, School of Health Sciences, University of Ioannina, 45110 Ioannina, Greece; 2Laboratory of Agronomy, Department of Crop Science, Agricultural University of Athens, 11855 Athens, Greece; 3Department of Pharmacy, Division of Pharmacognosy and Natural Products Chemistry, National and Kapodistrian University of Athens, Panepistimioupoli Zografou, 11527 Athens, Greece; 4Laboratory of Biological & Biotechnological Applications, Department of Agriculture, School of Agricultural Sciences, Hellenic Mediterranean University, Estavromenos, 71410 Heraklion, Greece; 5Institute of Agri-Food and Life Sciences, Research Center of the Hellenic Mediterranean University, Estavromenos, 71410 Heraklion, Greece; 6Laboratory of Chemistry, Department of Food Science & Human Nutrition, School of Food and Nutritional Sciences, Agricultural University of Athens, 11855 Athens, Greece; 7Symbeeosis S.A., 19002 Athens, Greece; 8Department of Biology, Faculty of Medicine, School of Health Sciences, University of Ioannina, 45110 Ioannina, Greece

**Keywords:** antioxidants, essential oils, hydroxytyrosol, oxidised low-density lipoprotein, dietary supplements

## Abstract

Several natural compounds have been explored as immune-boosting, antioxidant and anti-inflammatory dietary supplements. Amongst them, hydroxytyrosol, a natural antioxidant found in olive products, and endemic medicinal plants have attracted the scientific community’s and industry’s interest. We investigated the safety and biological activity of a standardised supplement containing 10 mg of hydroxytyrosol synthesized using genetically modified *Escherichia coli* strains and equal amounts (8.33 μL) of essential oils from *Origanum vulgare* subsp. *hirtum*, *Salvia fruticosa* and *Crithmum maritimum* in an open-label, single-arm, prospective clinical study. The supplement was given to 12 healthy subjects, aged 26–52, once a day for 8 weeks. Fasting blood was collected at three-time points (weeks 0, 8 and follow-up at 12) for analysis, which included full blood count and biochemical determination of lipid profile, glucose homeostasis and liver function panel. Specific biomarkers, namely homocysteine, oxLDL, catalase and total glutathione (GSH) were also studied. The supplement induced a significant reduction in glucose, homocysteine and oxLDL levels and was tolerated by the subjects who reported no side effects. Cholesterol, triglyceride levels and liver enzymes remained unaffected except for LDH. These data indicate the supplement’s safety and its potential health-beneficial effects against pathologic conditions linked to cardiovascular disease.

## 1. Introduction

Hydroxytyrosol (HT) is a bioactive compound with nutritional and pharmacological activities [1]. It is mainly characterised as (a) an antioxidant due to its capability to scavenge free radicals and activate endogenous antioxidant enzymes (and thus reduce oxidation of biological molecules in vivo), (b) an anti-inflammatory agent through the suppression of inflammatory markers and (c) an anti-atherogenic agent by inhibiting low-density lipoprotein (LDL) oxidation and platelet aggregation [2,3]. Several studies have documented the disease-preventive profile of HT either alone as a supplement or as part of a diet rich in olive oil [4,5]. The European Food and Safety Administration has adopted the claim that HT “protects LDL particles from oxidative damage” when more than 5 mg of HT and its derivatives (e.g., oleuropein complex and tyrosol) are consumed in olive oil daily by the general population [6]. Thus, HT constitutes an ideal candidate for the development of supplements and food additives (nutraceuticals). According to Commission Recommendation 97/618/EC3, hydroxytyrosol is allocated to Class 1.2, i.e., “foods and food components that are single chemically defined substances or mixtures of these which are not obtained from plants, animals or microorganisms that have been genetically modified and whose source has no history of food use in the Community”. Ιsolation of HT from plants is an expensive process [7] and chemically synthesised HT from precursor molecules has failed to yield high productivity [8]. Production of HT through metabolically engineered microbial strains at industrial or semi-industrial type bioreactors is an excellent alternative and environmentally friendly method that produces highly pure HT [9]. Intake of HT differs significantly between the countries of the European Union, which reflects the culture of each country towards a Mediterranean-style diet (high consumption of olive oil and olives). It is no surprise, therefore, that five Mediterranean countries occupy the first six places with Greece showing the highest HT intake (6.82 mg/day). On the other side, Austria has the lowest HT intake with 0.13 mg/day while the mean dietary intake in EU countries is 1.97 ± 2.62 mg/day [10]. Since HT intake is relatively low in many countries, the application of HT in functional foods and dietary supplements might be a plausible use.

Many essentials oils (EOs) are used in traditional medicines and are characterised by their antioxidant and antimicrobial properties [11] due to the presence of several bioactive compounds [12,13]. They are natural products (cosmetics, fragrances, etc.) that are well tolerated by the human body with minimum side effects and are widely accepted by consumers [14]. EOs are commonly used as natural alternatives to synthetic antioxidants. Their antioxidant and antimicrobial properties render them compounds of great importance for pharmaceutical, food, agricultural and health industries [15]. Currently, more than 3000 EOs are known [16]; however, the potential use of EOs as supplements/or nutraceutical additives has not yet been fully investigated. The main application of EOs in the food industry is as alternatives for food preservation. They have found fertile ground in the food industry due to the consumers’ awareness of the effects of synthetic compounds on human health [17]. In the agricultural industry, EOs are used as insecticides and herbicides, for the germination of seeds and as antibacterial and antifungal agents [15]. Finally, various cosmetics products (such as moisturizers, lotions and cleansers for skin care) consist of Eos, exploiting their active ingredients to enhance anti-inflammatory, antimicrobial and antioxidant activity [18].

In the present market for dietary supplements, several olive leaf extract products are available that are standardized in oleuropein (a bound form of HT—conjugated and glucosylated form). Based on bioavailability/metabolism aspects, HT is finally liberated and circulated in the organs (up to 25 mg of HT can be available per capsule). The same stands for oregano (*Origanum vulgare* subsp. *hirtum*) and sage (*Salvia fruticosa*) supplements and to a lesser extent for sea fennel (*Crithmum maritimum*). Regarding the three medicinal plants, most supplements originate from dried extracts. Essential oils supplements are usually in a form of a high-concentration solution than might require dilution. We developed a novel dietary supplement (Antiox-Plus) containing 10 mg of HT synthesized using genetically modified *Escherichia coli* strains and equal amounts (8.33 μL) of essential oils from *Origanum vulgare* subsp. *hirtum, Salvia fruticosa* and *Crithmum maritimum* and performed an open-label, single-arm, prospective clinical study on healthy human volunteers. Here, we report data regarding the effect of this oral supplement on biochemical profiles and most importantly on specific biomarkers such as homocysteine and oxidised low-density lipoprotein (oxLDL), which are found to be associated with pathologic conditions linked to cardiovascular disease (CVD).

## 2. Materials and Methods

### 2.1. Study Design

This was an open-label, single-arm, prospective clinical study designed [19] and conducted over 12 weeks. To evaluate the safety and biological activity of the supplement, 12 healthy volunteers were enrolled. The study was carried out at the Faculty of Medicine at the University of Ioannina. The participants were men and women, aged 26–52 years, from which written informed consent was obtained before the study initiation. Exclusion criteria were the presence of any chronic health conditions (diabetes, hypertension, dyslipidaemia), intake of nutritional supplements over the past 60 days, heavy smokers (≥25 cigarettes/day) and high alcohol use (men > 14 drinks/week, women > 7 drinks/week). The study was approved by the Research Ethics Committee at the University of Ioannina (ID 61742/2022) and was registered at http://clinicaltrials.gov under the number NCT05679310 (access date: 10 January 2023).

### 2.2. Intervention

The study was designed to evaluate the safety and biological activity of the supplement Antiox-Plus: one capsule/day, 15 min before their main meal, which contained 10 mg of HT, synthesised using genetically modified *Escherichia coli* strains [9] and equal amounts (8.33 μL) of essential oils from *Origanum vulgare* subsp. *hirtum, Salvia fruticosa* and *Crithmum maritimum* [20,21]. The volunteers consumed the supplement for 8 weeks. A follow-up analysis was performed one month after the end of the supplementation period (12 weeks from the initiation of the study). During this time, the participants consumed a placebo.

The volunteers were instructed to maintain their normal dietary habits. The following measurements were performed: (A) Dietary assessment at the beginning of the study (week 0), (B) Body composition analysis at the beginning (week 0) and the end of the study (week 8) and (C) biochemical and laboratory analysis of plasma samples at week 0, 8 and 12.

We asked the volunteers to record their food intake for 3 days (including one day of the weekend) and to fulfil a food frequency questionnaire. Analysis of their data was done by a certified nutritionist. Nutritional assessment was performed using the Evexis dietary software. The Tanita Dual Frequency Body Composition Monitor “Innerscan” was employed to measure weight, muscle mass, muscle quality score, heart rate, body fat (%), physique rating, visceral fat, metabolic age, basal metabolic rate (BMR), bone mass, body water (%) and body mass index (BMI) in the morning. Physical activity was assessed by the Greek version of the short International Physical Activity Questionnaire (IPAQ-short) [22].

### 2.3. Preparation of the Antiox-Plus Capsule

The Antiox-Plus capsule was prepared by Symbeeosis S.A. and contained (except for the essential oils and the liquid HT) Tween 80 and maltodextrin as suspending and filling agents, respectively. HT was isolated from *E. coli* strains as previously described [9] and identified using spectrometric (LC-HRMS) and spectroscopic methods (1 and 2D NMR) [7].

All metabolically engineered strains were tested for the presence of heterologous genes with diagnostic PCR according to Trantas et al. [9]. The essential oils were extracted from plants derived from oregano (*Origanum vulgare* subsp. *hirtum*), sage (*Salvia fruticosa*) and sea fennel (*Crithmum maritimum*) crops, where good farming practices were applied. Plant samples were harvested at the full blooming stage and air dried and stored at room temperature in the dark for 20 days, until their moisture content reached levels below 12%. Subsequently, the extraction of the essential oil was obtained by hydrodistillation. Specifically, dried leaves and inflorescences were pulverized, and 20 to 50 g of each species was submitted for hydrodistillation in a Clevenger apparatus for 4 h. Thereafter, the derived essential oils were collected, dried over anhydrous sodium sulphate and stored in dark bottles at −18 °C, until they were analyzed by Gas Chromatography-Mass Spectrometry (GC-MS). Analysis of the capsule was performed by an accredited laboratory and included the testing for methanol, cyclohexane, ethyl alcohol, ethylacetate, total aerobic microbial count, moulds and yeasts, enterobacteria, *Escherichia coli* and *Salmonella* spp. (certificate 2022-48E/22 007 139E, ERGANAL Lab, Piraeus, Greece).

### 2.4. Blood Count and Biochemical Analysis

Fasting blood was collected at three time points (weeks 0, 8 and 12) for biochemical and laboratory analysis. Biochemical analysis was done by a private microbiological-pathological laboratory and included: full blood count, fasting blood glucose (FBG), haemoglobin A1C (HbA1c), total cholesterol (TC), high-density lipoprotein cholesterol (HDL-C), low-density lipoprotein cholesterol (LDL-C), triglycerides (TG), serum glutamate-pyruvate oxaloacetic transaminase (SGPT), serum glutamic oxaloacetic transaminase (SGOT), gamma-glutamyltransferase (γ-GT), alkaline phosphatase (ALP), lactic dehydrogenase (LDH), total bilirubin (TBIL), direct bilirubin (DBIL), C-reactive protein (CRP) and homocysteine.

### 2.5. Laboratory Markers Analysis

Plasma oxLDL and interleukin-12 (IL-12) concentrations were measured by commercially available sandwich ELISA assay (Cusabio CSB-E07931h and Abcam ab46035). To examine the effect of the supplement on the glutathione concentration (GSH) and catalase activity, a glutathione assay kit (Sigma-Aldrich, CS0260) and a catalase assay (Megazyme, K-CATAL 04/20) were purchased. Lipid peroxidation (malonyldialehyde, MDA) in plasma specimens was determined by LC-MS (liquid chromatography-mass spectrometry) of the adduct obtained with thiobarbituric acid reagent [23], whereas estimation of total antioxidant capacity (TAC) was performed by blue CrO_5_ assay [24].

### 2.6. Statistical Analysis

Data are expressed as mean ± standard error of the mean (SEM). The statistical significance between data means at different time points was determined by the Friedman non-parametric test (SPSS version 20.0, SPSS Inc. Chicago, IL, USA). *p*-values < 0.05 were considered significant.

## 3. Results

Twenty-eight individuals were initially screened. Eleven were excluded for not meeting the inclusion criteria and five refused to participate. Twelve subjects (six women and six men) were included, and all completed the study. The mean age of the participants was 40.1 ± 9.7 years (40.7 ± 9.1 for the men and 39.5 ± 12.4 for the women). Five were non-smokers, four were light smokers (1–10 cigarettes/day) and three were moderate smokers (11–19 cigarettes/day) (Figure 1).

The average BMI was 25.9 kg/m^2^ even though subjects ranged from 19.6 to 30.6 kg/m^2^. To be precise, four subjects were categorised as healthy (three female and one male), six as overweight (two female and four male) and two as moderately obese (one female and one male). The subjects engaged mostly in moderate or high physical activity (9/12) and the majority were non- or light smokers (9/12) (Table 1). The diet of the participants can be characterised as a high-fat, adequate-protein, low-carb diet as seen in Table 2.

Figure 2 shows the changes in the lipid profile of the participants (Figure 2A–D) and glucose homeostasis (Figure 2E,F). A small but non-significant reduction in TC was noted at the end of the supplementation period (Figure 2A), mainly due to the reduction by 10% of triglycerides levels (Start 99.5 ± 19.7 mg/dL vs. End 87.8 ± 14.5 mg/dL) (Figure 1B). The changes in LDL-cholesterol and HDL-cholesterol were also non-significant with the former showing a reduction of 3% (Start 120.8 9.1 mg/dL vs. End 117.3 ± 8.6 mg/dL) and the latter an increase of 4% (Start 51.2 ± 2.7 mg/dL vs. End 53.0 ± 2.5 mg/dL) (Figure 2C,D). Initial levels of lipid profile were restored at follow-up (only HDL-cholesterol remained slightly elevated: Start 51.2 ± 2.7 mg/dL vs. Follow-up 54.0 ± 4.0). On the other hand, FBG was significantly reduced following the Antiox-Plus supplementation (Start 96.8 ± 1.5 mg/dL vs. End 92.8 ± 0.9 mg/dL, *p* < 0.05). The capsule produced an almost 5% decrease in blood glucose levels (Figure 2E). At follow-up, this significant effect on FBG levels was not observed in the participants (Start 96.8 ± 1.5 mg/dL vs. Follow up 95.6 ± 1.4 mg/dL). As expected HbA1c, a marker that changes more slowly remained stable (Figure 2F).

The liver function panel is presented in Figure 3A–G. No statistically significant variations were observed in the levels of SGOT, SGPT, g-GT, ALP, TBIL and DBIL (Figure 3A, Figure 3B, Figure 3C, Figure 3D, Figure 3F and Figure 3G, respectively) following 8-week supplementation with the Antiox-Plus capsule. However, a statistically significant (*p* < 0.05) decrease in LDH levels was recorded. Specifically, post-supplementation levels of LDH were reduced by 26% (Start 250 ± 17 U/L vs. End 184 ± 20 U/L) and were gradually increased thereafter without being restored completely to the initial levels (follow-up: 223 ± 16 U/L). CRP concentration remained within the normal range, although a significant increase was seen at the follow-up (Start 0.92 ± 0.46 mg/dL, End 0.73 ± 0.29 mg/dL, Follow up 2.2 ± 0.34 mg/dL) (Figure 3H).

Haematological analyses also showed that all parameters were within the normal range and no significant changes were observed following supplementation with the Antiox-Plus capsule (neither at the End nor at the Follow-up) (Table 3).

The impact of Antiox-Plus capsule supplementation on oxLDL levels was significant. OxLDL concentration reduced from 30.8 ± 2.4 mU/mL to 18.2 ± 0.9 mU/mL (*p* < 0.05). Cessation of the supplementation caused a 32% uprise of oxLDL levels in the plasma of the participants one month after (Follow-up); despite that, oxLDL levels remained 22% lower than baseline (Start 30.8 ± 2.4 mU/mL vs. 24.0 ± 1.1 mU/mL) (Figure 4A). No significant differences were observed in catalase activity or GSH concentration. Neither the mild fall in catalase activity at the end of the study (Start 35.5 ± 2.8 units/mL vs. End 29.3 ± 2.3 units/mL) nor its increase at follow-up (Start 35.5 ± 2.8 units/mL vs. Follow-up 40.8 ± 2.0 units/mL) was statistically significant alterations. GSH remained stable throughout the study period ranging from 820.0 ± 15.4 mU/mL at the beginning of the study to 790.5 ± 12.5 mU/mL at follow-up (Figure 4B,C). There were no considerable changes in the plasma lipid peroxidation as well as TAC of the participants during the study (Figure 4D,E).

Figure 5 shows the effect of the Antiox-Plus capsule on homocysteine levels. Interestingly, the supplement caused a significant reduction in homocysteine levels. Post-supplementation levels were lower by 2.6 μmol/L (Start 10.7 ± 0.81 μmol/L vs. End 8.1 ± 0.64 μmol/L, *p* < 0.05), which equals a 24% reduction. Henceforth, homocysteine gradually increased at 9.6 ± 0.69 μmol/L one month after the termination of supplementation.

## 4. Discussion

The present clinical trial evaluated the safety and the effects of a novel dietary supplement containing 10 mg of HT synthesized using genetically modified *Escherichia coli* strains and 25 μL of equal amounts (8.33 μL) of essential oils from *Origanum vulgare* subsp. *hirtum, Salvia fruticosa* and *Crithmum maritimum.* To our knowledge, this is the first study to evaluate a supplement originating from genetically modified *Escherichia coli* in healthy subjects. We found that daily intake of one capsule for 8 weeks significantly reduced the plasma levels of fasting blood glucose, homocysteine and oxLDL in 12 adults with a mean age of 40 years.

In a crossover study of 60 pre-hypertensive [systolic blood pressure (SBP): 121–140 mmHg; diastolic blood pressure (DBP): 81–90 mmHg] men, intake for 6 weeks of a phenolic-rich olive leaf extract (OLE), which contained 136 mg oleuropein and 6 mg of HT, led to a statistically significant reduction of TC, LDC-C and TG. Glucose metabolism on the other hand was not influenced by OLE. These findings indicate that OLE exerted lipid-lowering effects in vivo [25]. Lockyer et al. also showed a reduction in oxLDL concentration, but this decrease was statistically non-significant (Baseline 72 ± 24 U/L vs. End 69 ± 22 U/L, *p* = 0.124). Although in our protocol the supplementation period was longer (8 weeks vs. 6 weeks) with a greater amount of HT (10 mg/day vs. 6 mg/day) than the aforementioned study, we failed to record any effect on the lipid profile; however, we detected a decrease in oxLDL levels. We can speculate that the lipid-lowering potential of OLE is attributed to other phenolics and not only HT. The main component of olive leaves is the secoiridoid glucoside, oleuropein, which according to the evidence from in vivo studies is slowly transformed into its metabolic derivatives, hydroxytyrosol and elenolic acid [26]. A small number of studies have evaluated the effects of HT or essential oil supplementation outside food matrices in humans. Recently, a double-blinded, randomised, placebo-controlled crossover design study of 28 healthy volunteers (aged 18–65 years) showed that daily consumption of 15 mg HT in the form of gastrointestinal-resistant capsules for three weeks significantly improves oxidative stress plasma biomarkers (increase of total antioxidant status, SOD1 (superoxide dismutase 1) enzyme and thiol groups and reduction of nitrite, nitrate and MDA) [27]. HT was also detectable in the subjects’ blood (mean 2.83 μg/mL, min-max 2.25–3.50 μg/mL). Supplementation resulted in a significant upregulation of SOD1 gene expression (3.7-fold increase). No significant change was observed in catalase gene expression, which could explain why catalase activity remained stable in our study. In contrast with our findings, the authors reported an increase in LDL-C and no changes in glucose or oxLDL levels. The differences between the two studies could be attributed to the youngest population and the higher HT ingestion in a shorter time frame. Nonetheless, these differential outcomes might also indicate that supplementation with HT needs to be personalised to maximize its health benefits. Crespo et al. focused their research on the impact of HT supplementation on Phase II enzymes and found that a daily intake of 5 or 25 mg HT for seven days in the form of an olive mill wastewater extract (selectively enriched in HT) does not activate Phase II enzymes in the peripheral blood of mononuclear cells in 21 healthy volunteers [28]. Daily administration of HT at a dosage of 45 mg for 8 weeks in patients with mild hyperlipidaemia (total cholesterol values between 200 and 239 mg/dL) does not affect the markers of CVD, blood lipids, inflammatory markers, liver or kidney functions and the electrolyte balance but doubles endogenous vitamin C concentration, indicating an alternative physiologically antioxidant function for HT [29]. De Bock et al. showed that HT’s bioavailability and metabolism depend on several factors including the delivery method (capsule or liquid). In this small study (nine volunteers), ingestion of HT (and oleuropein) as a liquid preparation leads to peak levels of conjugated metabolites of HT 2.4-fold higher (61 ± 69 ng/mL versus 145 ± 154 ng/mL) than capsules and the area under the curve also 2.4-fold greater (5565 ± 4864 ng/mL versus 13,356 ± 11,678 ng/mL), even though HT concentration was higher in the capsules than in the liquid (9.7 versus 5.4 mg) [30]. The pharmacokinetics and bioavailability in humans are also affected by the food matrix in which HT is incorporated, with extra virgin olive oil being the best matrix for this compound [31]. Administration of a pure HT aqueous supplement (2.5 mg/kg body weight) in healthy volunteers leads to a fast absorption and detection of HT in the plasma in less than 15 min (13.5 ± 1.5 min). However, HT was completely undetectable after 2 h and was excreted mainly in conjugated forms (glucuronide or sulphate) in the urine 24 h after administration [32].

The biological properties of essential oils are generally described by their major components, which generally belong to two distinct groups: terpenes/terpenoids and aromatic/aliphatic constituents. The major components of oregano essential oil are carvacrol (14.5%), β-fenchyl alcohol (12.8%), thymol (12.6%) and γ-terpinene (11.6%) [33]; for sage essential oil, they are camphor (25.14%), α-thujone (18.83%), 1,8-cineole (14.14%), viridiflorol (7.98%), β-thujone (4.46%) and β-caryophyllene (3.30%) [34]; and for crithmum essential oil, they are *γ*-terpinene (33.6%), sabinene (32.0%) and thymol methyl ether (15.7%) [35]. Although EOs are studied as supplements for farm animals to improve meat quality, data on humans (from clinical studies) are scarce. In 2022, Maral et al. studied the effects of the essential oils of *Origanum dubium* (DUB), *Origanum vulgare* subsp. *hirtum* (HIR) and *Lavandula angustifolia* (LAV) on athletes’ lipid profiles and liver biomarkers. The subjects were divided into four groups (control, DUB, HIR and LAV) and received 2 mL EO three times a day (diluted in 150 mL of warm water after each meal) for two weeks. HDL-C significantly increased in DUB and HIR groups and LDL-C significantly decreased in the DUB group only [36]. Another study reported that treatment of patients with mild hyperlipidaemia by *Origanum onites* supplement (25 mL of the aqueous distillate of *Origanum onites* for three months) elicited a significant increase in HDL-C and a significant decrease in LDL-C compared to the controlled group [37]. We opted for a combination/synergistic effect from our mixture of EOs. However, we failed to observe any effects on HDL-C and LDL-C, which could be related to the relatively low dose of the supplement.

Hyperhomocysteinemia is a known risk factor for CVDs. Prolonged exposure to higher-than-normal levels of homocysteine can lead to the development of atherosclerosis and stroke [38]. Moreover, the oxidation of LDL particles—and thus the formation of oxLDL—can trigger inflammation through the activation of macrophages and other cells and lead to atherosclerosis [39]. Thus, keeping homocysteine and oxLDL concentrations at normal levels through the intake of vitamins or supplements might be a preventative approach to control inflammation and atherosclerosis-related conditions. Our study showed that the Antiox-Plus supplement significantly reduced homocysteine and oxLDL levels in healthy subjects. A previous study in healthy volunteers has shown that a significant amount of HT is present in the LDL-purified fractions 10 min after ingestion of an aqueous solution (2.5 mg/kg body weight) without affecting the plasma’s TAC or MDA levels [32]. This is in accordance with our findings and could offer a potential explanation for why the TAC in the plasma of the volunteers remained unaffected, whereas levels of oxLDL were reduced. However, to investigate the therapeutic effects of the Antiox-Plus supplement, further studies are required on people with hyperlipidaemia, obesity or metabolic syndrome. The literature research indicates data missing in clinical studies on HT or EOs supplementation. Characteristically, in the review article from Pastor et al. about the beneficial effects of dietary supplementation with olive oil, oleic acid or HT in metabolic syndrome, the authors identified only four eligible studies that met their search criteria [40]. The abundance of EOs also limits the extensive and in-depth investigation of specific EOs on specific health conditions and/or risk factors [41].

Our study had two main limitations. The first limitation was the small sample size. The COVID-19 pandemic limited the recruitment of patients and volunteers. The second limitation was the short duration of treatment. However, the supplement decreased risk factors correlated with CVD and did not cause any damage to the liver, as indicated by the stable levels of SGPT enzyme. Moreover, we found no effects of supplementation on SGOT, g-GT and ALP levels, which are used to predict damage to organs with high metabolic activity such as the muscles, kidneys, brain, heart, lungs, and liver.

## 5. Conclusions

Overall, for the first time, an open-label, single-arm, prospective clinical study on healthy human volunteers was performed to examine the safety and biological effects of the Antiox-Plus supplement, a mixture of HT synthesized using genetically modified *Escherichia coli* strains and EOs from *Origanum vulgare* subsp. *hirtum, Salvia fruticosa* and *Crithmum maritimum*. No adverse effects were reported by the participants during the supplementation period. Compliance with the Antiox-Plus supplement was generally high with more than 90% of the capsules provided being consumed by the subjects. The supplement was well tolerated by the volunteers. Elevated homocysteine levels are considered a risk factor for the development of acute ischemic myocardial disease and homocysteine blood levels in one of the most reliable diagnostic markers of CVD. oxLDL contributes to atherosclerotic plaque formation and the progression of plaque rupture can cause CVD. The daily intake of the Antiox-Plus capsule for 8 weeks exerted a favourable effect against homocysteine and oxLDL, and thus it is a promising supplementary approach for the reduction of biomarkers that are directly associated with pathologic conditions linked to CVD. Further clinical trials with larger samples and potentially in adults with risk factors for CVD could confirm the health benefits of Antiox-Plus supplementation.

## Figures and Tables

**Figure 1 microorganisms-11-00770-f001:**
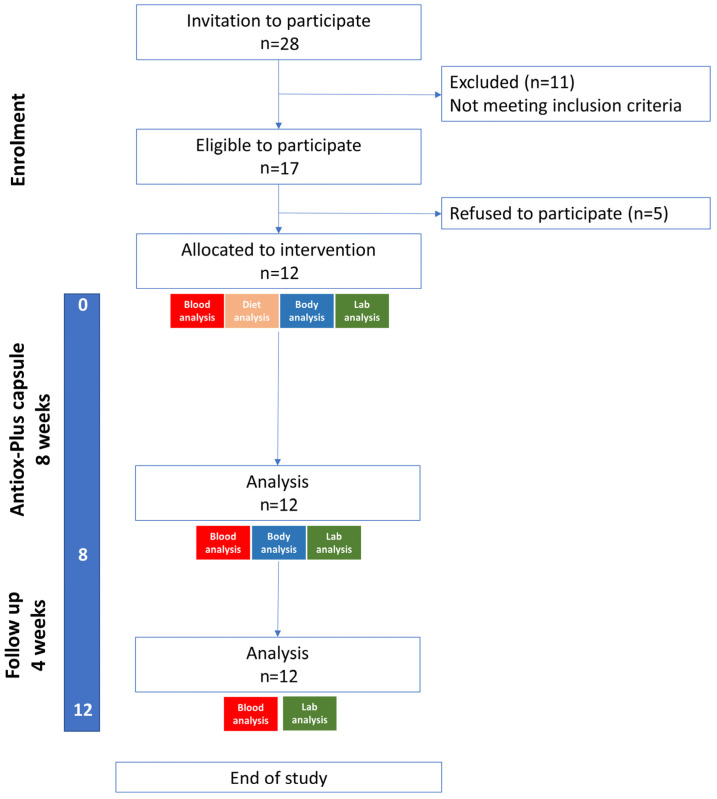
Study flow diagram.

**Figure 2 microorganisms-11-00770-f002:**
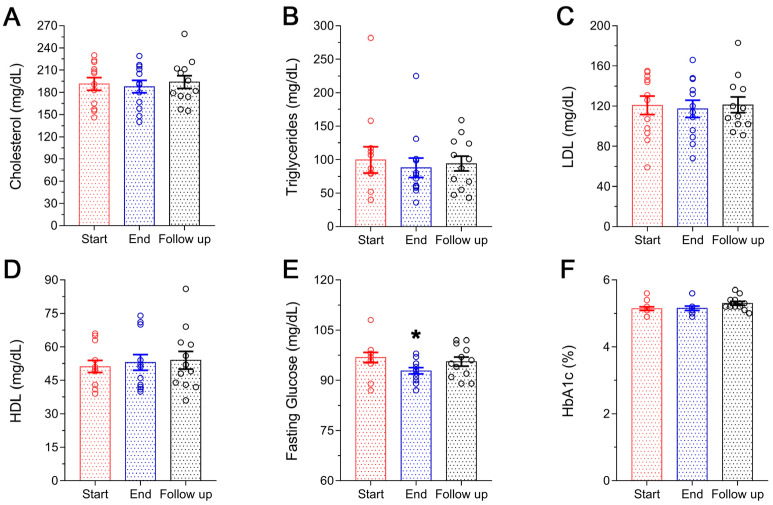
Lipid profile and glucose homeostasis of the subjects during the study. No significant changes were observed in the lipidemic profile of the volunteers following Antiox-Plus treatment (**A**–**D**). Serum levels of FBG were significantly decreased following Antiox-Plus supplementation (*p* < 0.05). Glucose levels were reset to pro-supplementation levels at follow-up (**E**). HbA1c levels remained stable throughout the study period (**F**). The dots represent individual measures in 12 blood samples from healthy volunteers each time. Statistically significant difference from Start (Day 0) is shown as * (*p* < 0.05).

**Figure 3 microorganisms-11-00770-f003:**
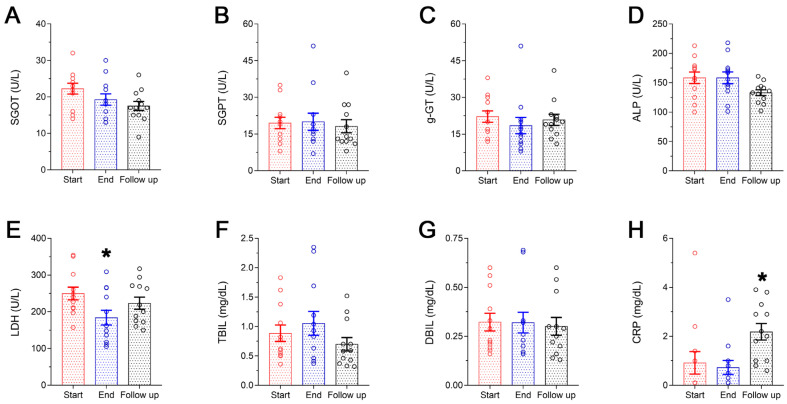
Liver function panel (**A**–**G**) and CRP levels (**H**) of the subjects during the study. A steep decrease in LDH was seen following Antiox-Plus supplementation (*p* < 0.05), which was restored to pro-supplementation levels at follow-up (**E**). Liver function was not affected by Antiox-Plus treatment. The dots represent individual measures in 12 blood samples from healthy volunteers each time. Statistically significant difference from Start (Day 0) is shown as * (*p* < 0.05).

**Figure 4 microorganisms-11-00770-f004:**
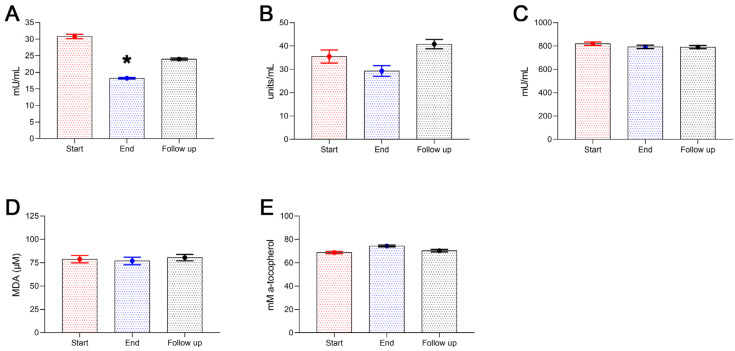
Antioxidant enzyme activity and markers of oxidative stress of the subjects during the study. Plasma levels of oxLDL significantly decreased following Antiox-Plus supplementation (*p* < 0.05) (**A**). oxLDL increased at follow-up, however, it remained lower than the pro-supplementation period. The Antiox-Plus supplement also exerted a non-significant reduction in catalase activity (**B**). Likewise, glutathione was unaffected by the supplement (**C**). The total antioxidant capacity of plasma was slightly increased (**D**), and lipid peroxidation levels remained stable (**E**). Statistically significant difference from Start (Day 0) is shown as * (*p* < 0.05).

**Figure 5 microorganisms-11-00770-f005:**
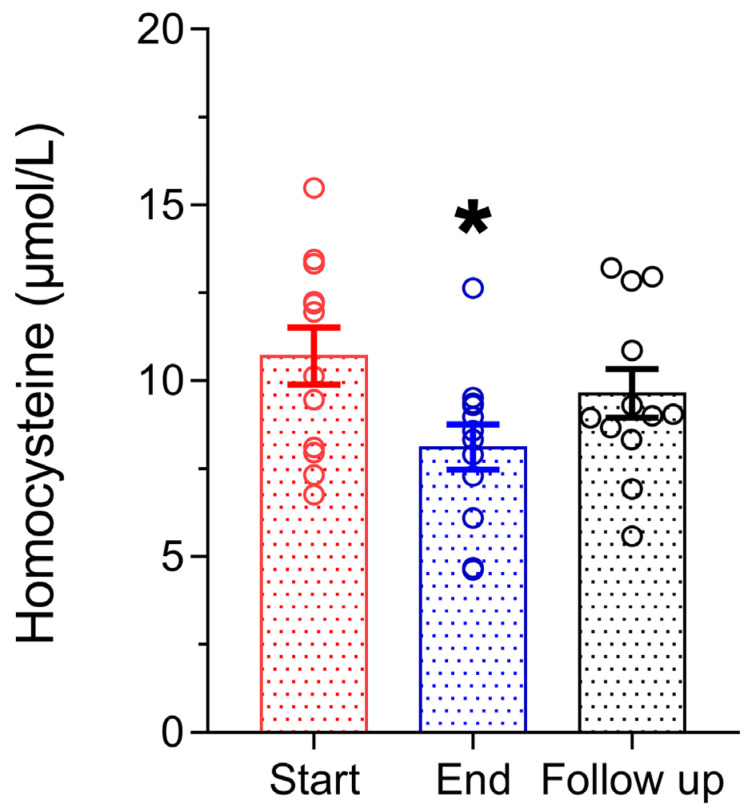
Homocysteine concentration in plasma was significantly decreased after Antiox-Plus treatment (*p* < 0.05). This effect was partially reset at follow-up, where homocysteine levels increased but remained lower than in the pro-supplementation period. Statistically significant difference from Start (Day 0) is shown as * (*p* < 0.05).

**Table 1 microorganisms-11-00770-t001:** Baseline characteristics of the participants.

Parameter		Mean	Female	Male
Weight (kg)		77.0 ± 13.6	68.1 ± 10.8	85.8 ± 10.2
Height (cm)		172 ± 8	166 ± 5	178 ± 6
ΒΜΙ (kg/m^2^)		25.9 ± 3.7	24.7 ± 4.1	27.1 ± 3.2
Fat mass (%)		29.7 ± 6.1	32.3 ± 7.4	26.9 ± 3.1
Muscle (kg)		51.4 ± 9.2	43.5 ± 2.9	59.4 ± 5.0
Total body water (%)		51.4 ± 4.0	49.4 ± 4.4	53.3 ± 2.6
Physical activity (Kcal/week)	Low	3/12	1/6	2/6
	Moderate	5/12	3/6	2/6
	High	4/12	2/6	2/6
Smoking	No	5/12	1/6	4/6
	Light	4/12	3/6	1/6
	Moderate	3/12	2/6	1/6
BMR (kcal)		1613 ± 279	1381 ± 102	1846 ± 176

**Table 2 microorganisms-11-00770-t002:** Study participants’ macronutrient assessment.

		Mean	Female	Male
Energy	(Kcal/day)	2185 ± 220	2100 ± 260	2300 ± 110
Carbohydrates	%	44	47	40
Proteins	%	15	14	16
Lipids	%	41	39	44

**Table 3 microorganisms-11-00770-t003:** Changes in the blood count of the participants during the study.

	Reference Range	Start	End	Follow-Up
White blood cells (WBC) (10^3^/μL)	4.0–10.0	6.7 ± 0.4	6.0 ± 0.3	6.3 ± 0.3
Red Blood Cells (RBC) (10^6^/μL)	4.2–6.0	5.0 ± 0.2	4.8 ± 0.2	4.8 ± 0.2
Haemoglobin (HGB) (g/mL)	13.5–18.0	14.4 ± 0.4	13.8 ± 0.4	13.7 ± 0.3
Haematocrit (HCT) (%)	40.0–52.0	43.4 ± 1.2	41.3 ± 1.1	40.8 ± 1.0
Platelets (10^3^/mL)	140–440	250 ± 19	237 ± 17	246 ± 20

## Data Availability

The data presented in this study are available on request from the corresponding author. The data are not publicly available due to ethical restrictions.

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
