# Peer review of "Oral Supplementation with Hydroxytyrosol Synthesized Using Genetically Modified Escherichia coli Strains and Essential Oils Mixture: A Pilot Study on the Safety and Biological Activity"

_microorganisms, 2023, doi:10.3390/microorganisms11030770_

Round 1
Reviewer 1 Report
The study conducted by Yannis et al., is on the “Oral supplementation with hydroxytyrosol synthesized using 2 genetically modified Escherichia coli strains and essential oils 3 mixture: a pilot study on the safety and biological activity”, is well written and can be a new oral supplementation used against cardiovascular diseases. Due to many shortcomings in the manuscript. I suggest considering this as short communication instead of full article.
1. Why the authors use a mixture of oil obtained from three plants (Origanum vulgare, Salvia officinalis and Crithmum maritimum) instead of one. I think, I would be better if the authors performed this study by using the oil of each plant separately and then to performed biological activities.
2. The authors did not mention that the oils were extracted from these plants or purchased from the market. If the authors used commercially available oil, then the activity may be due to the other residues instead of pure oil.
3. How the authors identified genetically modified Escherichia coli strains
4. How did the authors identify the compound (hydroxytyrosol) in the strain. As NMR is basic tool used for identification which is not mentioned in the manuscript.
5. The essential oil contains a lot of components, especially one or two compounds in a major quantity, which can be responsible for the reduction of these activities.
6. The authors used 10 mg hydroxytyrosol synthesized using genetically modified Escherichia coli strains and tested for patients. I think would be if different concentrations were used with different quantity of essential oils to see the effect with a minimum amount of these ingredients.
7. The number of volunteers is 12 which is very less number for the conclusion and not enough without composition of these oils and quantifying the amount of hydroxytyrosol in the strains.
8. Table 1 and 2 is not according to the journal style.
Author Response
1. Why the authors use a mixture of oil obtained from three plants (Origanum vulgare, Salvia officinalis and Crithmum maritimum) instead of one. I think, I would be better if the authors performed this study by using the oil of each plant separately and then to performed biological activities.
Our study is part of a project funded by Greece and the European Union (Please see funding information). We have investigated the in vitro (cell study) and in vivo (mice) effects of these substances (our manuscript is under review after passing the first review round with minor recommendations). Our decision on the composition of the Antiox-plus capsule was based on three points: 1) our knowledge obtained by the in vitro and in vivo experiments, 2) the composition of the commercially available supplements and 3) the taste test performed by the company (Symbeeosis SA).
2. The authors did not mention that the oils were extracted from these plants or purchased from the market. If the authors used commercially available oil, then the activity may be due to the other residues instead of pure oil.
The essential oils were extracted from these plants. The procedure was performed by the Agricultural University of Athens. The following text was added to the revised manuscript: “The essential oils were extracted from plants derived from, oregano (Origanum vulgare subsp. hirtum), sage (Salvia fruticosa) and sea fennel (Crithmum maritimum) crops, where good farming practices were applied. Plant samples were harvested at the full blooming stage, were air dried and stored, at room temperature in the dark for 20 days, until their moisture content reached levels below 12%. Subsequently, the extraction of the essential oil was obtained by hydrodistillation. Specifically, dried leaves and inflorescences were pulverized and 20 to 50 g of each species was submitted to hydrodistillation in a Clevenger apparatus for 4 hours. Thereafter, the derived essential oils were collected, dried over anhydrous sodium sulphate and stored in dark bottles at -18 oC, until they were analyzed by Gas Chromatography – Mass Spectrometry (GC-MS)”.
3.How the authors identified genetically modified Escherichia coli strains
The following text was added to the revised manuscript: “All metabolically engineered strains were tested for the presence of heterologous genes with diagnostic PCR according to [Trantas, E.; Navakoudis, E.; Pavlidis, T.; Nikou, T.; Halabalaki, M.; Skaltsounis, L.; Ververidis, F. Dual Pathway for Metabolic Engineering of Escherichia Coli to Produce the Highly Valuable Hydroxytyrosol. PLoS One 2019, 14, e0212243, doi:10.1371/JOURNAL.PONE.0212243]”.
4. How did the authors identify the compound (hydroxytyrosol) in the strain. As NMR is basic tool used for identification which is not mentioned in the manuscript.
The following text was added to the revised manuscript: “HT was isolated from E. coli strains as previously described and identified using spectrometric (LC-HRMS) and spectroscopic methods (1 & 2D NMR) [Angelis, A.; Hamzaoui, M.; Aligiannis, N.; Nikou, T.; Michailidis, D.; Gerolimatos, P.; Termentzi, A.; Hubert, J.; Halabalaki, M.; Renault, J.H.; et al. An Integrated Process for the Recovery of High Added-Value Compounds from Olive Oil Using Solid Support Free Liquid-Liquid Extraction and Chromatography Techniques. J. Chromatogr. A 2017, 1491, 126–136, doi:10.1016/J.CHROMA.2017.02.046]”.
5. The essential oil contains a lot of components, especially one or two compounds in a major quantity, which can be responsible for the reduction of these activities.
We agree with the reviewer’s comment.
6. The authors used 10 mg hydroxytyrosol synthesized using genetically modified Escherichia coli strains and tested for patients. I think would be if different concentrations were used with different quantity of essential oils to see the effect with a minimum amount of these ingredients.
There could be several combinations regarding the amount of HT and the essentials oils in the final capsule. As we mention before (please see answer for the first comment) our decision on the composition of the Antiox-plus capsule was based on three points: 1) our knowledge obtained by the in vitro and in vivo experiments, 2) the composition of the commercially available supplements and 3) the taste test performed by the company (Symbeeosis SA).
7. The number of volunteers is 12 which is very less number for the conclusion and not enough without composition of these oils and quantifying the amount of hydroxytyrosol in the strains.
Due to the COVID-19 pandemic, we faced difficulties in the enrolment of volunteers for the study. Nonetheless, we were able to include 12 volunteers which is the minimum number recommended for pilot studies (usually referred to as a “rule of 12”-see reference below). Our primary target was to estimate the average values of biochemical markers after the supplementation with the Antiox-Plus capsule. Moreover, this study was a preparatory investigation to provide us with information to plan larger-scale clinical studies.
* Moore CG, Carter RE, Nietert PJ, Stewart PW. Recommendations for planning pilot studies in clinical and translational research. Clin Transl Sci. 2011;4(5):332-7.
8. Table 1 and 2 is not according to the journal style.
Tables 1 and 2 were formatted according to the journal’s style.
Reviewer 2 Report
1- The title is suitable and it does not need any changes.
2- Abstract should improved, the material and methods part of the Abstract should be improved, the section of result part of the Abstract is not complete it, but the final conclusion part is OK.
3- The Keywords are OK and it does not need any changes.
4- In Introduction, and other parts of the article, please, start each section and paragraph according to the new subject and context, not randomly.
5- The arrangement of paragraphs should be improved especially in Introduction part.
6- Materials and Methods and figures and tables are OK , and no changes are needed in this part.
7- The section of discussion is too short and it should be improved.
8- Like Discussion part, conclusion section should be improved, the conclusion should be a brief part of each manuscript, it needs improvement.
9- 35 References for one Article is not enough, that is why you Discussion section is not appropriate and it is not enough which needs to be improved.
All in all the article just needs Minor revision, and after minor revision, it can be accepted for publication.
Author Response
1- The title is suitable and it does not need any changes.
Thank you for your comment.
2- Abstract should improved, the material and methods part of the Abstract should be improved, the section of result part of the Abstract is not complete it, but the final conclusion part is OK.
The abstract was modified according to the reviewer’s comments. The material and methods and the results part were expanded to offer a more clear view of the study to the reader.,
3- The Keywords are OK and it does not need any changes.
Thank you for your comment.
4- In Introduction, and other parts of the article, please, start each section and paragraph according to the new subject and context, not randomly.
Thank you for your comment. We rearranged the paragraphs in the Introduction section to include three main sections a) hydroxytyrosol, b) essential oils and c) the current market for these supplements and the aim of the study.
5- The arrangement of paragraphs should be improved especially in Introduction part.
Paragraphs were rearranged in the introduction part.
6- Materials and Methods and figures and tables are OK , and no changes are needed in this part.
Thank you for your comment.
7- The section of discussion is too short and it should be improved.
The discussion section was extended.
8- Like Discussion part, conclusion section should be improved, the conclusion should be a brief part of each manuscript, it needs improvement.
The conclusion section was modified.
9- 35 References for one Article is not enough, that is why you Discussion section is not appropriate and it is not enough which needs to be improved.
The discussion section was extended, and the total number of references was increased to 43
All in all the article just needs Minor revision, and after minor revision, it can be accepted for publication.
Thank you for your comment.
Reviewer 3 Report
The authors aimed to develop a novel dietary supplement (Antiox-Plus) containing 10 mg of HT synthesized using genetically modified Escherichia coli strains and equal amounts (8.33 μL) of essential oils from Origanum vulgare, Salvia officinalis and Crithmum maritimum and performed an open-label, single-arm, prospective clinical study on healthy human volunteers. In this study they report data regarding the effect of this oral supplement on the biochemical profile and most importantly on specific biomarkers such as homocysteine and oxidised low-density lipoprotein (oxLDL) that are found to be associated with pathologic conditions linked to cardiovascular disease.
The study covers some issues that have been overlooked in other similar topics. The structure of the manuscript appears adequate and well divided in the sections. Moreover, the study is easy to follow, but some issues should be improved. Some of the comments that would improve the overall quality of the study are:
I-) Authors must pay attention to the technical terms acronyms they used in the text;
II-) Please better stated the limitation of the study;
III-) Conclusion Section: This paragraph required a general revision to eliminate redundant sentences and to add some "take-home message".
Author Response
I-) Authors must pay attention to the technical terms acronyms they used in the text;
Thank you for your comment. Indeed, we spotted some mistakes in our technical terms acronyms which were corrected in the revised manuscript.
II-) Please better stated the limitation of the study;
The limitations of the study were moved to the discussion section in a separate paragraph.
III-) Conclusion Section: This paragraph required a general revision to eliminate redundant sentences and to add some "take-home message".
The conclusion section was rewritten.
Round 2
Reviewer 1 Report
I don't feel this work to be published in a high impact factor journal with the current research data.
Author Response
I don't feel this work to be published in a high-impact factor journal with the current research data.
We were surprised to read the reviewer’s 1 comment/decision on the revised version of our manuscript. We have studied reviewer’s 1 constructive comments carefully and have revised the manuscript following his/her suggestions. We included a point-by-point response to the reviewer’s 1 comments. The Reviewer’s 1 decision to degrade the quality of our work in the second round of review was unexpected and contradicts his/her initial judgment.